# The Impact of the Main Negative Socio-Economic Factors on Female Fertility

**DOI:** 10.3390/healthcare10040734

**Published:** 2022-04-14

**Authors:** Viorel Țarcă, Elena Țarcă, Florin-Alexandru Luca

**Affiliations:** 1Faculty of Communication Sciences, Apollonia University, Păcurari Street No. 11, 700511 Iassy, Romania; vtarca@gmail.com; 2Department of Surgery II-Pediatric Surgery, “Grigore T. Popa” University of Medicine and Pharmacy, Universității Street No. 18, 700115 Iassy, Romania; 3Department BMTM, “Gheorghe Asachi” Technical University, Bulevardul Profesor Dimitrie Mangeron 67, 700050 Iaşi, Romania; florin.alexandru.luca@gmail.com

**Keywords:** crude birth rate, income level, alcohol consumption, tobacco consumption, body mass index

## Abstract

The negative relationship between fertility and income is well known to economists and demographers. Developed countries have experienced a remarkable decline in their fertility rate as they have become richer. Lifestyle choices can affect a woman’s ability to conceive. Tobacco use and heavy drinking is associated with an increased risk of ovulation disorders, and being overweight or significantly underweight can inhibit normal ovulation. Our research is focused on evaluating the main risk factors that influence female fertility. We assembled a country-specific dataset on birth rate and socio-economic factors for 171 countries, using data integrated from publicly available data sources. The regression model shows that the negative factor with the greatest impact on female fertility is represented by the level of income per capita. The negative effects of smoking, alcohol consumption, and body weight on female fertility are also demonstrated, but with a lower impact compared to the average income per capita.

## 1. Introduction

A continuous decline in fertility and low birth rates appear to be widespread problems worldwide, but the consequences are more dramatic in developed countries, due to the much lower level of fertility and the implications for simple reproduction in the context of the complex process of demographic transition. For example, the birth rate in the European Union (EU) decreased from 16.4 live births per 1000 persons in 1970 to 12.8‰ in 1985, 10.5‰ in 2000, and 9.3‰ in 2019 [1]. As we can easily observe, the birth rate in the EU decreased at a slower pace between 2000 and 2019 than before, but this decreasing trend remains an important problem to solve in the future. The year 2020 was the first year in which the population of the EU-27 saw a slight decline, ending a steady but less and less intense increase recorded until then. In a situation of continuous decline in fertility and steadily rising mortality, the natural growth in the EU-27 population has become negative since 2011, the total change, however, has remained positive only due to net migration. EU-27 member states are in the higher per capita income categories of the World Bank classification, being positioned in stages 4 and 5 (or between them) of the demographic transition process, characterized by the fact that the population starts to decline due to the birth rate falling below the death rate, as the population is no longer replacing itself. At the other end of the spectrum are the least developed countries, respectively in the first two stages of the demographic transition, initially characterized by a rapid growth rate of the population, followed by a tendency to reduce the mortality rate as a result of a slight improvement in living conditions and health care.

With regard to fertility, the World Population Data Sheet 2021 released annually by the Population Reference Bureau highlights differences based on income levels and age. In high-income countries, fertility rates for women of all age groups have declined since 1950. Fertility rates are also lowest for adolescents aged 15–19 and women in their 40s (14 and 8 births per 1000 women, respectively). In middle-income countries, the fertility rate for women in their 30s has stabilized, which the Data Sheet suggests could signal a transition to delayed childbearing in middle-income countries, as seen in high-income countries. In low-income countries, fertility rates are steadily declining for all age groups, but without signs of signal-delayed childbearing. The fertility rate for adolescent’s ages 15 to 19 in these countries is 94 births per 1000 adolescent girls. The global total fertility rate (lifetime number of births per woman) is 2.3, which is above replacement-level (2.1 births per woman) but lower than it was in 1990 (3.2) [2].

### Aim of the Study

The purpose of this study is to investigate the relationship between socio-economic factors and female fertility as measured using birth rate. We aim to measure this at the country-level to simultaneously assess the effects of multiple socio-demographic factors from around the world on female fertility (i.e., crude birth rate).

## 2. Materials and Methods

### 2.1. Sources of Information

In this study we used the 2021 World Bank classification of world’s economies in four income groups, using the Atlas method exchange rates of the previous year: low-gross national income (GNI) per capita in current USD < 1045 USD, lower-middle (1046–4095 USD), upper-middle (4096–12,695 USD), and high (>12,695 USD) income countries [3].

We assembled a country-specific dataset on birth rate and socio-economic factors using data integrated from publicly available data sources.

-For female fertility, we used the country-specific birth rate per 1000 population as a proxy. To obtain birth rate data, we used data collected and maintained by the World Health Organization (WHO) for 2019 [4].-For the mean body mass index (age-standardized estimate, in kg/m^2^) we also used data collected and maintained by the World Health Organization in 2016 [5].-For the average consumption of pure alcohol in liters per year among all adults aged 15+ (in liters of pure alcohol for all beer, wine, spirits, and other alcoholic drinks) in 2019 we used data collected and maintained by World Population Review [6].-For the number of cigarettes smoked per person in 2016 (all adults aged 15+) we used data collected and maintained by The Tobacco Atlas, American Cancer Society, Inc. and Vital Strategies [7].-Finally, for the income level per capita we used data collected and maintained by World Bank [8].

Overall, our dataset contained information on five variables: crude birth rate, income level, female mean body mass index, female average consumption of pure alcohol per capita per year, and average number of cigarettes smoked per person per day from 171 countries. This dataset contains about 78.8% (171/217) of states from around the world, for which all data on the studied indicators were available. Currently, the regions in the World Bank country database include 217 economies at all income levels, but for the indicators considered in the present study, there were not enough data available for 46 countries. The term country used in the World Bank classification does not imply political independence, but refers to any territory for which authorities report separate social or economic statistics.

### 2.2. Definitions

-The crude birth rate (CBR) measures the frequency or intensity of births in a population over a given period of time, usually a year, and is calculated by relating the number of live births to the average population in that period. The crude birth rate is the ratio of the number of births during the year to the average population in that year; the value is expressed per 1000 inhabitants [9].-The total fertility rate (TFR) is the mean number of children that would be born alive to a woman during her lifetime if she were to pass through her childbearing years conforming to the age-specific fertility rates of a given year. A total fertility rate of around 2.1 live births per woman is considered to be the replacement level in developed countries: in other words, the average number of live births per woman required to keep the population size constant in the absence of migration [1].-According to the WHO, Body Mass Index (BMI) is a person’s weight in kilograms divided by the square of height in meters and is classified as: normal-weight (18.5–24.9), overweight (25–29.9), class I obesity (30–34.9), class II obesity (BMI 35–39.9), and class III obesity (BMI ≥ 40) [10].-Gross national income (GNI) per capita (formerly GNP per capita) is the gross national income, converted to U.S. dollars using the World Bank Atlas method, divided by the midyear population. GNI is the sum of value added by all resident producers plus any product taxes (less subsidies) not included in the valuation of output plus net receipts of primary income (compensation of employees and property income) from abroad [8].

### 2.3. Methodology–Statistical Analysis

Data are expressed as mean, standard deviation, and minimum and maximum values. All calculations were made using standard statistical package (JASP Team (2022). JASP (Version 0.16.1), University of Amsterdam, The Netherlands, https://jasp-stats.org/).

Using available country-specific data from 171 countries, first we compared the means of the three independent variables (mean BMI, average consumption of pure alcohol, and average number of cigarettes smoked per person) for each income group, using unifactorial ANOVA or the non-parametric Kruskal–Wallis test, as appropriate. To find precisely which mean differs from another mean and if the difference between each pair of means is statistically significant, we used post hoc tests (Tukey’s HSD or Steel–Dwass–Critchlow–Fligner pairwise ranking nonparametric method). Correlations between the variables were investigated by the Kendall’s tau non-parametric correlation coefficient or by the Spearman rank correlation coefficient, as appropriate.

Finally, we used multiple linear regression analysis (stepwise method) to investigate the prognostic value of the independent variables score for predicting the dependent variable—crude birth rate. Our model assesses the negative impact of income level per capita, female mean BMI, female average consumption of pure alcohol in liters per year (age ≥ 15) and the average number of cigarettes smoked per person per day (age ≥ 15) on female fertility at the country-level. We used birth rate as a proxy for female fertility to enable us identify the level at which these factors negatively affect female fertility. The contribution of covariates explaining the dependent variable was assessed with a *p*-value of < 0.05 considered as significant.

In the regression analysis, the variable income level was transformed into a dummy variable, which records the value “0” for countries with average per capita income below the global median level in 2019 (according to our calculations, median GNI per capita in current USD in 2019 was 6180) and the value “1” for countries with an average per capita income value above the median value.

To verify a possible multicollinearity of the model, we analyzed variance inflation factors (VIF), eigenvalues of the scaled, condition indexes, and the variance proportions. The serial correlations between errors (autocorrelation assumption) were tested with the Durbin–Watson test. In the end, we checked if the errors are normally distributed, in other words if the differences between the model and the observed data are most frequently zero or very close to zero.

Regarding sample size, our data contain 171 cases, without any missing observation, so the minimum sample size condition was met.

## 3. Results

In order to validate research hypotheses (the relationship between socioeconomic factors and crude birth rate), comparisons among income groups and bivariate correlation analysis between the independent variables were in the first instance used in the current paper. Comparisons among groups were made by *p*-value (one-way analysis of variance for normally distributed data or Kruskal–Wallis test for non-normally distributed data).

We start by presenting the general situation of the three independent variables (female alcohol consumption per capita, mean female BMI, and cigarette consumption per day) for the 171 countries included in the study (Table 1).

Below, we present in detail the situation of the same three independent variables grouped by major income categories (Table 2), according to the World Bank 2021 classification (low income, lower-middle income, upper-middle income, and high-income countries, based on GNI per capita in current USD).

Of the three variables analyzed, only female mean BMI follows approximately a normal distribution, which is why in order to compare the world countries over the four income groups we used unifactorial ANOVA (Fisher), and the result was F(3.167) = 22.755, *p* < 0.001. For female alcohol consumption per capita and cigarette consumption per capita, variables that do not follow normal distributions, we used the non-parametric Kruskal–Wallis test (H(3) = 59.398 for female alcohol consumption per capita and H(3) = 37.382 for cigarette consumption, both with *p* < 0.001). Comparing the averages of the three independent variables for each income group, we immediately notice that at least two categories differ significantly in terms of the average values of the three independent variables studied (*p* < 0.001 for each variable).

Next, to see which pairs differed significantly, we had to carry out post hoc tests to compare all income groups with each other for all three independent variables in the model. For each pair of groups, we only added significant differences to the table (*p* < 0.01).

There are statistically significant differences in the behavior of individuals (age ≥ 15) relative to alcohol and cigarette consumption between countries with different income levels. Thus, in general, the average individual consumption of cigarettes and alcohol is higher as the level of per capita income between countries increases. The differences presented in Table 2, regarding the average annual levels of a female’s average annual alcohol consumption, are statistically significant, highlighting an important increase in the consumption as the income level increases. There is an increase of almost three times between the average alcohol consumption of a woman in the case of countries with a high-income level and those that fall into the lower middle-income category. Approximately the same situation is present in the case of a person’s daily cigarette consumption, from about one cigarette per day in the countries with the lowest income per capita, to almost four cigarettes in the countries with the highest incomes. An interesting and slightly different situation is signaled in the case of the average female weight levels (expressed by the BMI index). Thus, the BMI index registers slightly increasing levels in the case of the countries included in the first three income categories, from 23.3 in the “low income” countries with up to 27.3 in the situation of “upper middle income” countries. In the case of “high income” countries, after exceeding the value of 12,695 USD of GNI per capita, we notice a decrease in the average weight of females by more than 1 kg/s qm.

Next, to verify the correlation between the dependent variable (Birth rate) and each of the three predictors, for all the four income groups considered, we used Spearman’s or Kendall’s (only for the “low-income” small data set) correlation coefficients, because our data violated parametric assumptions as non-normally distributed data. In the “low-income” countries case we observe a negative correlation of the crude birth rate with BMI-female, and for “lower middle income” countries also a negative correlation with the average amount of cigarettes smoked in a day by one person, both statistically significant. For countries in higher income categories, the gross birth rate is negatively correlated with cigarette and alcohol consumption and positively correlated with BMI Female (all statistically significant).

In almost all countries, regardless of the category of income group, there is a tendency to decrease the birth rate as daily cigarette consumption increases, only that in the case of low-income countries, the correlation is weaker and not statistically significant. Also, from the analysis of the previous table, in the case of the average daily consumption of cigarettes per capita over the age of 15, the negative influence on the birth rate is stronger and stronger as the income level is higher (Table 3).

These aspects are very well highlighted by the following graphic representations (Figure 1, Figure 2 and Figure 3).

Regarding the influence of women’s alcohol consumption on female fertility, there is a statistically significant negative influence of the increase in alcohol consumption for middle- and high-income countries and the absence of any correlation in the situation of low-income countries, probably due to extremely low alcohol consumption and cigarettes that appear in official statistical reports (official bans on their consumption, unfavorable climate, lack of material possibilities to purchase them, and statistical reports with low fidelity).

The influence of mean female body weight (highlighted by the average BMI index) on female fertility takes place in opposite directions for the four major categories of countries by income level. Thus, in the case of the countries from the two upper-income categories, we observe a direct and intense correlation of BMI on the crude birth rate, in the case of the countries with the lowest incomes a negative correlation of medium intensity, while in the case of small- to medium-income countries we notice the absence of any correlation.

### Regression Model

The dependent variable (Crude birth rate) is quantitative and each independent variable used in the model is also quantitative or dichotomous. The variable “Income level” was transformed into a dummy variable, which records the value 0 for countries with average per capita income below the global average level in 2019 (median GNI per capita in current USD = 6180) and the value 1 for countries with an average per capita income value above the median value.

Regarding sample size, Samuel B. Green (1991) suggests two rules of thumb for the minimum acceptable sample size, the first to test the overall fit of the regression model (R^2^ value), and the second based on testing the individual predictors within the model (b-values of the model). In this case, to test the model overall, the recommended minimum sample size is 50 + 8 × k = 50 + 8 × 4 = 82 and to test the individual predictors, Green suggests a minimum sample size of 104 + k = 108 (where k is the number of predictors, in our study 3) [11]. Our data contain 171 cases, without any missing observation, so the minimum sample size condition was met (171 > 104).

First, we needed to use a matrix of the correlation coefficients for all the variables in the model and also mark the significance value of each correlation (* *p* < 0.05, ** *p* < 0.01 and *** *p* < 0.001). It is immediately noticeable that the crude birth rate is negatively related to income level (−0.670), cigarette consumption per day (−0.480), female alcohol consumption per capita (−0.548), and female mean BMI (−0.318), the significance level being less than 0.01 for all (Table 4). This significance value tells us that the probability of obtaining a correlation coefficient this big in a sample of 171 countries if the null hypothesis were true (there was no relationship between these variables) is very low.

A multiple linear regression was carried out to ascertain the extent to which income levels (bellow and above median), mean female BMI, female alcohol consumption per capita, and cigarette consumption per day per capita can predict the dependent variable—crude birth rate (Table 5). For performing the regression analysis, using the stepwise method, we used an initial model that contains only the constant and then the computer searches for the predictor (out of the ones available) that best predicts the outcome variable (has the highest simple correlation with the dependent variable). If this predictor significantly improves the ability of the model to predict the outcome, then this predictor is retained in the model and the computer searches for a second predictor (the variable that has the largest semi-partial correlation with the outcome) and so on. Each time a predictor is added to the equation, a removal test is made of the least useful predictor.

If the first version of our model contains only the first predictor (income level) that has the highest simple correlation with the dependent variable, by using the stepwise method, the final model contains four independent variables, valid for the crude birth rate prediction: income level, cigarette consumption per day, female mean BMI, and female alcohol consumption per capita per year. The fourth model presented in our study had a value of 0.777 for the multiple regression coefficients (R), which grew steadily since the introduction of each predictor, starting with the level of income per capita and ending with female alcohol consumption per capita, in the fourth step. When we used only the level of income per capita as a predictor, the simple correlation with the crude birth rate, our dependent variable was 0.670, as it appeared previously in the matrix of correlation coefficients (Table 4). To measure how much of the variability in the outcome is accounted for by the predictors, we used the value of R^2^. For the first model its value was 0.449, which means that level of income per capita accounts for 44.9% of the variation in crude birth rate. When the other three predictors were also included in the model, this value increased to 0.604 or 60.4% of the variance in crude birth rate. Therefore, if the level of income per capita accounts for 44.9%, we can tell that cigarette consumption per day, female mean BMI, and female alcohol consumption per capita account for an additional 15.5% of the variance in crude birth rate.

The fourth version of the model is the best, having an R of 0.777, R^2^ of 0.604, adjusted R^2^ of 0.594, and a standard estimated error of 6.130. As it can be observed, the value of R (0.777) proves a strong correlation between the level of fertility and the four independent variables (income level, cigarette consumption per day, mean female BMI, and female alcohol consumption per capita per year). The values of R^2^ (0.604) and adjusted R^2^ (0.594) indicate a high proportion of variation in the dependent variable explained by the regression model (about 60%).

The difference between adjusted R^2^ and R^2^ shows us how well our model generalizes. In our final model, the difference between the values is 0.01 (about 1%), meaning that if the model were derived from the entire world population rather than a sample (171 countries in our case), it would account for approximately 1% less variance in the outcome. To check the cross-validity predictive power of the model or how the statistical analysis results will generalize to an independent data set, we used the Stein formula [12]:(1)$$R_{c}^{2}=1-\left(\frac{n-1}{n-k-1}\right)\left(\frac{n-2}{n-k-2}\right)\left(\frac{n+1}{n}\right)\left(1-R^{2}\right),$$
where n is sample size (171), k is the number of predictors (4), and R^2^ is the unadjusted value (0.604).

For our data:(2)$$R_{c}^{2}=1-\left(\frac{171-1}{171-4-1}\right)\left(\frac{171-2}{171-4-2}\right)\left(\frac{171+1}{171}\right)\left(1-0.604\right)=0.582,$$

The value of $R_{c}^{2}$ is very close to the observed value of R^2^ (0.604), indicating that the cross-validity of our model is very good. Next, in Table 6 are presented the results of the ANOVA for the regression model (Table 6), which tests whether the model is a significant fit of the data overall (*p* value less than 0.05).

For all the models from one to four, the *p* values are 0, therefore the dependent variable (birth rate) is explained through the action of the independent variables. The fourth model is the most appropriate one. For this model, the analysis indicates that the sum of squares for regression is higher than the sum of squared residuals (9495.202 > 6236.942). Therefore, the model explains an important part of the variation in the dependent variable. Still, the value of the test significance F (*p* = 0.000 < 0.01) indicates that the independent variables largely explain the variation in the dependent variable.

The *p* values associated to the regression coefficients for the explicative variables: income level, cigarette consumption per day, female mean BMI, and female alcohol consumption per capita per year are lower than the significance level 0.05 considered, which means that the estimated coefficients are significant for the world population (Table 7).

Therefore, the equation of the regression model is the following:(3)$$\left(\begin{array}{c} \mathrm{Crude}\\ \mathrm{Birth}\\ \mathrm{Rate} \end{array}\right)=49.658-8.645\left(\begin{array}{c} \mathrm{Income}\\ \mathrm{level} \end{array}\right)-1.064\left(\begin{array}{c} \mathrm{Cigarette}\\ \mathrm{consumption}\\ \mathrm{per day} \end{array}\right)-0.780\left(\begin{array}{c} \mathrm{Female}\\ \mathrm{mean}\\ \mathrm{BMI} \end{array}\right)-1.028\left(\begin{array}{c} \mathrm{Female}\\ \mathrm{alcohol}\\ \mathrm{consumption}\\ \mathrm{per year} \end{array}\right),$$

Each B-coefficient indicates the average decrease in birth rate associated with a one-unit increase in a predictor, because they are all negative. Thus, for income level (which is “0” for countries below median level and “1” for those above), the only possible one-unit increase is from below “0” to above “1”. Therefore, B = −8.645 simply means that the average birth rate for countries above the median income is lower with 8.645% compared to the birth rate in countries below the median income. On the other hand, one extra cigarette smoked a day reduces the gross birth rate by 1%. The increase in mean female BMI by one-unit results in a decrease in the crude birth rate by approximately 0.8%. Finally, an increase of 1 L in the average amount of pure alcohol consumed annually by a woman (aged 15 and over) results in a 1% decrease in the crude birth rate, an effect relatively similar to that of an extra cigarette smoked per day by a person (male or female) aged 15 and over, as we have seen before. The variation of the crude birth rate is negatively affected by all four predictors considered in the model. Testing multicollinearity is important for checking correlations between independent variables of the regression model.

Tolerance is a useful indicator in testing the multicollinearity of independent variables. Multicollinearity is indicated by tolerance values close to 0. Since we do not have this situation for the variables of the model presented (all tolerance values are above 0.643), we can state that there is no multicollinearity between variables. In the case of VIF (variance inflation factor), a multicollinearity problem would be signaled by values of over 5 or even 10. Simple arithmetic mean of the four VIF’s calculated for our model is equal to 1.372, a value that is very close to 1 and once again confirms that collinearity is not a problem for our model.

Other measures that are useful in discovering whether predictors are dependent are the eigenvalues of the scaled, uncentred cross-products matrix; the condition indexes; and the variance proportions. The variance proportions vary between 0 and 1, and for each predictor it should be distributed across different dimensions (or eigenvalues). For our model, we can see that each predictor has most of its variance loading onto a different dimension (income has 48% of variance of the regression coefficient on dimension 2, cigarette consumption has 83% of variance on dimension 3, female BMI has 99% of variance on dimension 5, and female alcohol consumption has 89% of variance on dimension 4).

In the case of condition indices, values over 15, which may suggest collinearity issues, are on dimension 5 (condition index = 33.787). Still, variance proportions do not indicate multicollinearity issues, because only one predictor (female BMI) has 99% of variance on dimension 5, while the other three predictors have most of their variance distributed on another dimension.

The serial correlations between errors (autocorrelation assumption) were tested with the Durbin–Watson test.

From the output we can see that the test statistic is 1.998 (very close to 2) and the corresponding *p*-value is 0.996. Since this *p*-value is far above 0.05, we cannot reject the null hypothesis, according to which there is no correlation among the residuals. To test the normality of residuals, we must analyze the histogram and normal Q-Q plot of residuals (Figure 4).

In the case of our model, the histogram of standardized residuals looks like a normal distribution (almost a bell-shaped curve) and the Quantile−Quantile plot of the standardized residuals is almost a straight line, suggesting that the residuals are normally distributed. Thus, the multiple regression model does not seem to violate the normality assumptions significantly.

## 4. Discussion

There is a wide variation in multiple birth rates and trends over time in Europe and around the globe, with no straightforward pattern between geographical areas [13,14]. Socio-demographic and policy factors differ between countries and can impact birth rates. In developing countries, the rate of infertility is estimated at one in four couples [15]. A study from 2017 on 195 countries and territories shows that despite reductions in the TFR, the global population has been increasing by an average of 83.8 million people per year since 1985, reaching 7.6 billion people in 2017 [16]. However, 33 countries had a negative population growth rate from 2010 to 2017, most of which were located in Central, Eastern, and Western Europe. Because of the reduction in the TFR and the negative population growth rate, one can expect a massive migration of population from South Asia and Sub-Saharan Africa towards Europe in the next decades. In the same direction, one can expect a generalized increase in live births from foreign-born mothers in most of the EU countries. More than 65% of the children born in Luxembourg in 2019 were from foreign-born mothers, followed by Liechtenstein, Switzerland, Cyprus, Austria, and Belgium [1]. In the United States of America, between 12 and 16% of reproductive age women (15–44 years) suffer from impaired fecundity, and the European countries show an even higher infertility rates, with Great Britain at 17% [17,18]. We analyzed 171 countries around the globe to detect the most important socio-economic factors with negative impact on fertility and birth rates. Our regression model showed that the negative factor with the greatest impact on female fertility is represented by the level of income per capita: as societies progress, fertility tends to decrease. Thus, exceeding the median level of income per capita (6180 USD per year) by a certain country, initially situated below this level, leads to a decrease in the crude birth rate by about 8.65%. We could say that starting with a level of about $ 515 average monthly income for a person, every year the crude birth rate in a country or area would decrease by about 9 children for every 1000 inhabitants. Possible explanations are the opportunity costs of childbearing; investments in the quality instead of the quantity of children; and in the case of the second demographic transition, rising higher-order needs conflicting with childbearing aspirations [19,20].

A vast literature has documented the negative relationship between income or development and fertility, but there are recent, new theoretical considerations that suggest that at high levels of development the relationship between development and fertility might turn positive [19]. While global national-level comparisons demonstrate negative relationships between fertility and development levels, recent evidence suggests that in the future, among high-income countries, this relationship may transition from negative to positive [19,21]. This we think remains to be explored because the robustness and interpretation of these findings continue to be debated. Another study found that for OECD (Organization for Economic Co-operation and Development) member countries, economic development is likely to induce a fertility re-increase in the richest societies, but this increase will be small if driven by increase in Gross Domestic Product (GDP) per capita only. The authors suggest that GDP per capita has to reach US$66.000 for fertility to increase back to replacement level (2.1 children per women) [22]. However, this is very unlikely to happen, because even today GDP per capita level is still far below that level.

The extent to which environments and communities are supportive and enabling is fundamental in shaping the behaviors of individuals. Since the obesity in adults is a problem in many countries we looked at, this will increase the rate of obesity in children as well, and even worsen the health problems of future generations. The prevention of child and adolescent overweight and obesity will rely on helping people to eat healthy foods and to engage in regular physical activity, including by ensuring that these are accessible, available, and affordable options [10]. This will require the engagement of multiple sectors, including education, communications, commerce, urban planning, agriculture, and health [10]. Overweight and obesity are major problems of public health, being causal factors of several diseases and also intervening at different levels of the reproductive function. However, data from the literature are contradictory in terms of the effect of obesity and its severity on live birth rate. There are studies that have not found a deleterious effect of BMI, with no significant difference in the live birth rate between normal-weight and overweight women, or between normal-weight and obese women [23]. Also, women obesity does not impact the cumulative live birth rate after in vitro fertilization [24]. However, there are also studies that have shown a negative impact of overweight and of each of the grades of obesity on the live birth rate compared to normal BMI [25]. Female obesity is often associated with impaired spontaneous fertility and adverse pregnancy outcomes [26]. A recent meta-analysis demonstrated that live birth rate is significantly decreased in obese women compared with normal-weight women, with a relative risk of 0.85 (CI 95%: 0.84–0.87) [27]. These controversial results could be explained by the differences in study populations, as most published studies were conducted in the US, with different inclusion criteria. Our study is important because it analyzed 171 countries all around the globe and the regression model found that in the case of female mean BMI the increase by one unit of the index determines the reduction by 0.8% of the crude birth rate. The BMI index registers slightly increasing levels in the case of the countries included in the first three income categories, from 23.3 in the “low income” countries with up to 27.3 in the situation of “upper middle income” countries, while in the case of “high income” countries we notice a decrease in the average weight of females by more than 1 kg/s qm. Very interesting is the influence of mean female body weight on female fertility, which takes place in opposite directions for the four major categories of countries by income level. Thus, in the case of the countries from the two upper-income categories, we observe a direct and intense correlation of BMI on the crude birth rate, in the case of the countries with the lowest incomes a negative correlation of medium intensity, while in the case of small- to medium-income countries we notice the absence of any correlation.

Few studies have analyzed the relationship between BMI, alcohol, and tobacco use, and birth rate. Tobacco smoking and alcohol abuse have serious consequences on both individual and public health [28,29]. Both tobacco smoking and alcohol abuse increase the risk of illicit drug abuse in the young population and health problems occur more frequently and with more adverse consequences in polydrug-using persons [30]. Although it seems that there is a declining trend in smoking worldwide, the prevalence of tobacco smoking is still high in women in the general population (30% in France, 25% in Germany, 16% in the United Kingdom, and 15% in the United States of America) [31], and the prevalence of smoking among women of reproductive age has increased [32]. We demonstrated that an extra cigarette smoked per day by a person (male or female) aged 15 and over, regardless of the level of income per capita in that country, results in a 1% decrease in the crude birth rate. We also observed statistically significant differences in the behavior of individuals (age ≥ 15) relative to cigarette consumption between countries with different income levels. Thus, in general, the average individual consumption of cigarettes is higher as the level of per-capita income between countries increases. In almost all countries, regardless of the category of income group, there is a tendency to decrease the birth rate as daily cigarette consumption increases. The analysis shows that the negative influence of cigarettes consumption on the birth rate is stronger as the income level is higher.

Tobacco use during pregnancy is particularly dangerous, as smoking is linked to adverse health outcomes for both the mother and the developing child [33]. Due to the fact that cigarette smoke contains more than 4000 harmful substances, tobacco smoking is one of the greatest risk factors of more than 60% noncommunicable diseases and is also implicated in decreased male and female fertility [34,35]. Parental tobacco exposure in utero was reported to affect male fertility because of the direct exposure of male offspring during the embryonic developmental processes [35]; therefore, tobacco affects not only the fertility of the present generation, but also that of the future generation. Reducing maternal smoking could lessen the occurrence of infant SGA (small for gestational age) and decrease socioeconomic inequalities in birth weight for GA (gestational age) [36].

Besides tobacco use, alcohol use is among the most significant risk factors for burden of disease. The prevalence of alcohol use and related harms is between 2 and 12 times higher in men than women, but there is recent evidence to suggest that this gap is closing among recently born cohorts [37,38]. This suggests that young women in particular should be the target of concerted efforts to reduce the impact of substance use and related harms [37,39]. Rates of alcohol use among women are different, depending on cultural customs of each country, but women’s alcohol excess may affect their sexual behavior and is associated with the risk of unplanned pregnancy. Ethanol consumption (5%) 2 weeks before pregnancy results in a decrease in the number of viable fetuses and abnormal fetal development, and these effects are accompanied by impaired maternal glucose homeostasis and hepatic steatosis during pregnancy [40]. Anyway, little is yet known about the exact influence of maternal drinking before pregnancy on fetal development and growth. Ethanol has a bad impact on all tissues, but growing and developing ones are the most susceptible to alcohol due to incomplete antioxidant protection, therefore, stem cell and progenitor cell function could be affected [41]. The increasing rates of alcohol consumption in women and in the prenatal period are serious social and health problems, but there a very few studies analyzing the effect of alcohol on birth rates. As we demonstrated, an increase of 1 L in the average amount of pure alcohol consumed annually by a woman (aged 15 and over) results in a 1% decrease in the crude birth rate.

As in the case of cigarette smoking, we observed an important increase in the alcohol consumption as the income level increases. We also noticed an increase of almost three times between the average alcohol consumption of a woman in the case of countries with a high-income level and those that fall into the lower middle-income category. Our study indicates a statistically significant negative influence of the increase in alcohol consumption for middle- and high-income countries and also the absence of any correlation in the situation of low-income countries, probably due to extremely low alcohol consumption and cigarettes that appear in official statistical reports. Public health policy must act for the reduction or prevention of drinking before pregnancy, to ensure maternal and fetal health during all phases of pregnancy and after birth.

There are studies confirming the role of potentially modifiable population factors such as BMI, smoking, alcohol, and environmental exposures in determining preterm birth risk for example, but not in birth rate also. Our final model indicates a high proportion of variation in the dependent variable explained by the regression equation (60.4%). Within this variation, the level of income per capita accounts for 44.9% and cigarette consumption per day; female mean BMI and female alcohol consumption per capita play account for an additional 15.5%. These negative factors interact and are associated with more general health and social and economic policies that promote healthy childbearing, as we demonstrated. More knowledge about how these factors will contribute to birth rates would be enormously useful for shaping future policy.

### Limitations of the Study

There are several limitations to this study. First, we performed a global analysis investigating each country equally. However, not all countries are the same size. We used statistics such as birth rate per 1000 population, income level (as a dichotomous variable), mean body mass index trends among adults (age-standardized), average consumption of pure alcohol in liters per year (age ≥ 15), and the average number of cigarettes smoked per person per day (age ≥ 15), which should help to address some of these issues. However, there are some fundamental differences between large and small countries that are not fully addressed by our methods. Therefore, this remains a limitation of our approach. We only investigated variables that have been reported to impact female fecundity and fertility and that had data available at the country-level for the all of countries included in our study. Therefore, not every factor implicated in female fertility was included in our model. Some of the major negative factors (income level per capita, average body mass index, average number of cigarettes smoked, and average alcohol consumption per capita) were included.

The best way to measure how birth rates are changing is not the CBR but the TFR. This measure provides an age-controlled estimate of how many kids a woman beginning her childbearing years now would have over her whole life if current birth rates remain stable. Due to the available data, we had to use in our analysis the crude birth rate.

The World Bank’s classification of world economies into four income groups for 2021 was used, and the data for the indicators in the model refer to 2016 and 2019, depending on availability.

## 5. Conclusions

Presenting results based on recent data can be a valuable addition to the literature because populations adapt to socio-demographic and economic factors. Our final model indicates a high proportion of variation in the dependent variable explained by the regression equation (60.4%). Within this variation, level of income per capita accounts for 44.9% and cigarette consumption per day; female mean BMI and female alcohol consumption per capita account for an additional 15.5%.

Our regression model shows that the negative factor with the greatest impact on female fertility is represented by the level of income per capita. Thus, exceeding the median level of income per capita (6180 USD per year) by a certain country, initially situated below this level, leads to a decrease in the crude birth rate by about 8.65%. In other words, we could say that starting with a level of about $515 average monthly income for a person, every year the crude birth rate in a country or area, which is now below median income level, would decrease by about 9 children for every 1000 inhabitants. An increase of 1 L in the average amount of pure alcohol consumed annually by a woman (aged 15 and over) results in a 1% decrease in the crude birth rate, an effect relatively similar to that of an extra cigarette smoked per day by a person (male or female) aged 15 and over, if the effects of all other predictors are held constant. A slightly lower impact was observed in the case of female mean BMI, where the increase by one unit of the index determined the reduction by 0.8% of the crude birth rate.

We also checked the cross-validity of our model and the assumptions of regression, so we can probably assume that it is both accurate for the sample used and generalizable to the entire world population.

## Figures and Tables

**Figure 1 healthcare-10-00734-f001:**
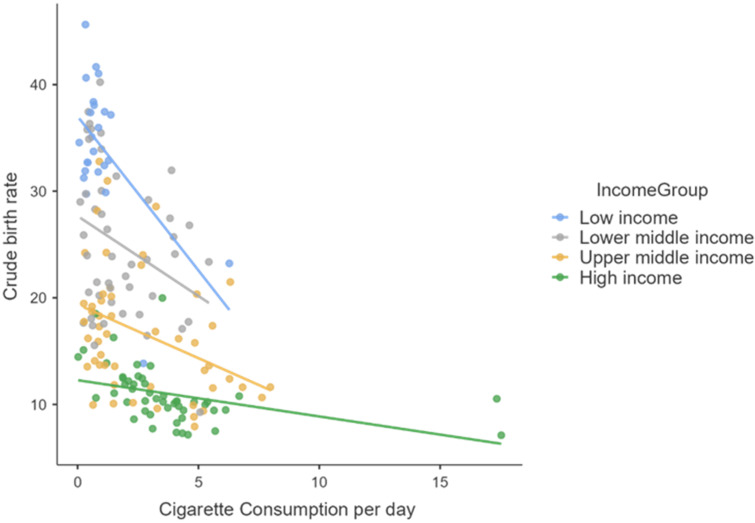
The correlation between the number of cigarettes consumed daily per capita and the crude birth rate.

**Figure 2 healthcare-10-00734-f002:**
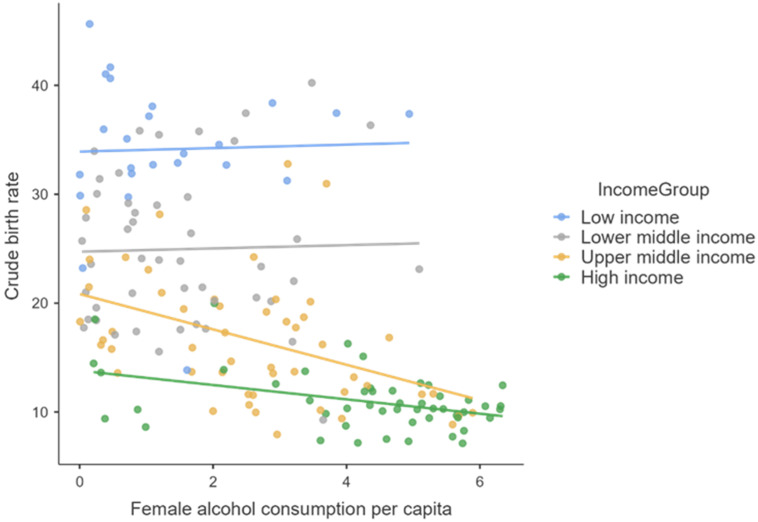
The correlation between female’s alcohol consumption per capita and the crude birth rate.

**Figure 3 healthcare-10-00734-f003:**
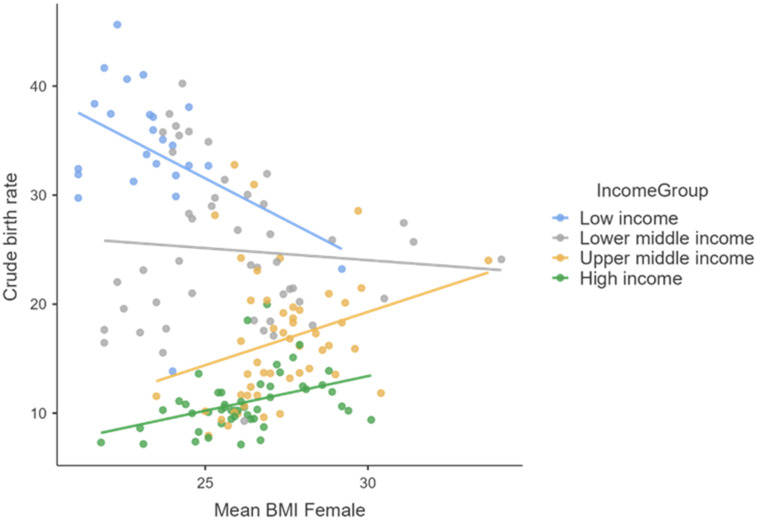
The correlation between mean BMI Female and the crude birth rate.

**Figure 4 healthcare-10-00734-f004:**
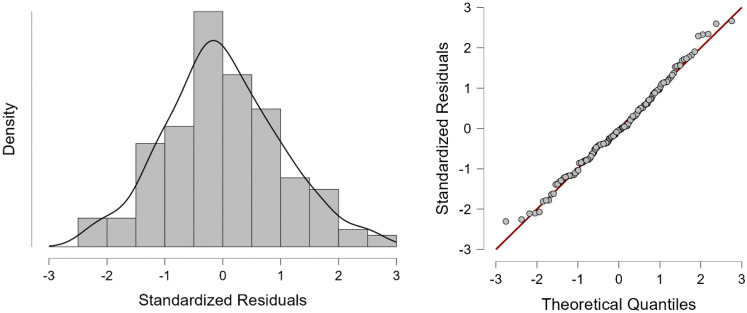
Histogram and normal Q−Q plot of standardized residuals.

**Table 1 healthcare-10-00734-t001:** General situation of the independent variables.

Variable	N		Mean	Standard Deviation	Min.	Max.
	Valid	Missing				
Female alcohol consumption per capita (15+) 2019	171	0	2.584	1.889	0.003	6.340
Mean female BMI (kg/m^2^) 2016	171	0	26.042	2.317	21.100	34.100
Cigarette Consumption per person per day 2016	171	0	2.522	2.482	0.027	17.530

**Table 2 healthcare-10-00734-t002:** Situation of the independent variables grouped by major income categories.

Income Group	Mean				Standard Deviation			
	Low Income	Lower Middle Income	Upper Middle Income	High Income	Low Income	Lower Middle Income	Upper Middle Income	High Income
N	24	47	51	49	24	47	51	49
Female alcohol consumption per capita (15+) 2019	1.326	1.505	2.564	4.254	1.284	1.243	1.580	1.727
Mean female BMI (kg/m^2^) 2016	23.321	25.855	27.325	26.216	1.694	2.566	1.675	1.699
Cigarette Consumption per day 2016	1.001	1.810	2.684	3.780	1.247	1.510	2.239	3.219

**Table 3 healthcare-10-00734-t003:** Bivariate correlations between birthrate and each independent variable.

Income Group	N	Correlation	Cigarette Consumption per Day	Female Alcohol Consumption per Capita	Mean BMI Female
Low income	24	Kendall’s tau	−0.109	0.004	**−0.261 ***
		*Sig. (1-tailed)*	*0.228*	*0.490*	*0.039*
Lower middle income	47	Spearman’s rho	**−0.296 ***	0.010	−0.023
		*Sig. (1-tailed)*	*0.022*	*0.474*	*0.439*
Upper middle income	51	Spearman’s rho	**−0.393 ****	**−0.464 ****	**0.376 ****
		*Sig. (1-tailed)*	*0.002*	*0.000*	*0.003*
High income	49	Spearman’s rho	**−0.679 ****	**−0.294 ***	**0.462 ****
		*Sig. (1-tailed)*	*0.000*	*0.020*	*0.000*

Bold text highlights significant values. ** Correlation is significant at the 0.01 level (1-tailed). * Correlation is significant at the 0.05 level (1-tailed).

**Table 4 healthcare-10-00734-t004:** Correlation matrix for all the variables in the model.

Variable	Crude Birth Rate 2019	Income Level 2019	Cigarette Consumption per Day 2016	Female Alcohol Consumption per Capita (15+) 2019	Mean BMI Female (kg/mp) 2016
Crude birth rate 2019	1.000				
Income level 2019	−0.670 ***	1.000			
Cigarette Consumption per day 2016	−0.480 **	0.240 **	1.000		
Female alcohol consumption per capita (15+) 2019	−0.548 **	0.546 ***	0.378 ***	1.000	
Female mean BMI (kg/mp) 2016	−0.318 **	0.230 **	0.112	−0.021	1.000

Note. ** *p* < 0.01, *** *p* < 0.001.

**Table 5 healthcare-10-00734-t005:** Summary of the estimated regression model.

Model ^e^	R	R Square	Adjusted R Square	Std. Error of the Estimate	R Square Change
1	0.670 ^a^	0.449	0.445	7.165	0.449
2	0.746 ^b^	0.557	0.551	6.444	0.108
3	0.761 ^c^	0.579	0.571	6.299	0.022
4	0.777 ^d^	0.604	0.594	6.130	0.025

^a^ Predictors: (Constant), income level. ^b^ Predictors: (constant), income level, cigarette consumption per day. ^c^ Predictors: (constant), income level, cigarette consumption per day, female mean BMI. ^d^ Predictors: (constant), income level, cigarette consumption per day, female mean BMI, female alcohol consumption per capita. ^e^ Dependent Variable: Birth Rate 2019.

**Table 6 healthcare-10-00734-t006:** ANOVA summary of the estimated regression model.

ANOVA ^a^						
Model		Sum of Squares	df	Mean Square	F	Sig.
1	Regression	7057.165	1	7057.165	137.483	0.000 ^b^
	Residual	8674.979	169	51.331		
	Total	15,732.144	170			
2	Regression	8755.645	2	4377.822	105.422	0.000 ^c^
	Residual	6976.499	168	41.527		
	Total	15,732.144	170			
3	Regression	9104.982	3	3034.994	76.480	0.000 ^d^
	Residual	6627.162	167	39.684		
	Total	15,732.144	170			
4	Regression	9495.202	4	2373.800	63.180	0.000 ^e^
	Residual	6236.942	166	37.572		
	Total	15,732.144	170			

^a^ Dependent variable: birth rate 2019. ^b^ Predictors: (constant), income level. ^c^ Predictors: (constant), income level, cigarette consumption per day. ^d^ Predictors: (constant), income level, cigarette consumption per day, female mean BMI. ^e^ Predictors: (constant), income level, cigarette consumption per day, female mean BMI, female alcohol consumption per capita.

**Table 7 healthcare-10-00734-t007:** Coefficients of the estimated regression model.

Model		Unstandardized Coefficients		Standardized Coefficients	t	Sig.	95.0% Confidence Interval for B	
		B	Std. Error	Beta			Lower Bound	Upper Bound
4	(Constant)	49.658	5.568		8.918	0.000	38.664	60.651
	Income Level 2019	−8.645	1.169	−0.451	−7.395	0.000	−10.953	−6.337
	Cigarette consumption per day 2016	−1.064	0.206	−0.274	−5.154	0.000	−1.471	−0.656
	Female mean BMI 2016	−0.780	0.213	−0.188	−3.652	0.000	−1.201	−0.358
	Female alcohol consumption per capita per year 2019	−1.028	0.319	−0.202	−3.223	0.002	−1.657	−0.398

Dependent Variable: Birth Rate 2019.

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
