# Peer review of "The Impact of the Main Negative Socio-Economic Factors on Female Fertility"

_healthcare, 2022, doi:10.3390/healthcare10040734_

Round 1
Reviewer 1 Report
The authors have examined the relationship between socio-economic factors and female fertility measured using birth rate. In addition, the authors also measure this relationship at the country-level to assess the effects of multiple socio-demographic factors from around the world on female fertility or the crude birth rate. Overall, the authors have examined the research question in detail and provided a lot of analysis. The authors provide a lot of details about the statistical analysis methods and provide reasonings in support of their methods. Although, it is a well written manuscript, the authors should consider some changes to the manuscript especially in the results section. The manuscript is very long and trimming details would make it easier to read and follow.
A comment about the numbers in the manuscript: all numbers which should have a decimal seem to have a comma. This change needs to be made throughout.
Here are detailed comments for different sections of the manuscript:
Introduction
- Provides a good background and appropriate statistics.
- The objectives are well mentioned.
- It might be a better idea to move the section 1.2 “sources of information” to the methods section. The section should probably end after the aims.
- The first line under “sources of information” needs a reference when the authors mention the 2021 World Bank classification.
Methods
- The section provides good definitions for the factors examined.
- How were the socio-economic factors selected? What is the rationale for selecting BMI, alcohol intake, and smoking?
- Rationale for selecting a sample size of 171 cases are given in the results section. It would be more appropriate to move it to the methods section.
Results
- The authors provide a lot of detail in this section and at some places hard to follow. They should probably consider trimming this section and only include a few tables with key findings and probably include the other results in the supplementary section.
- It is difficult to follow Table 3 and not sure if it can just be mentioned in the text.
- It is not very clear why the authors have decided to show Table 4. They should consider mentioning it in the text or moving it to the supplementary section.
- Section 3.1 “Regression Models” is very detailed. Should consider trimming.
- Not sure why the authors have decided to include a correlation matrix in Table 6.
- Table 8 should be re-formatted and not sure if details about sum of squares, df, and mean square are needed.
- Table 10 might fit well in the supplementary section. The authors could probably briefly mention the findings from the multicollinearity tests.
- Not sure if Table 11 on diagnostics is needed in the main paper.
- Not clear why Table 12 is needed. The authors can mention this in the text.
- Figure 4 showing histogram and residual plots are probably not needed or can be moved to the supplementary section.
- Overall, the results section needs a lot of trimming and re-formatting. Presently, has too many details and some sections contains a lot of information. It might be better to have few tables and figures which show the key findings as it will make it easier to understand the findings.
Discussion
- This section should also be trimmed. Contains too many details.
- Should begin with a summary of findings and then move on to mentioning statistics from other countries.
- Well written section but with a little re-formatting and trimming, will become easier to follow.
Author Response
Response to Reviewer 1 Comments
Dear Reviewer,
Thank you very much for evaluating our manuscript. Your recommendations and comments have helped us greatly improve our manuscript. Here we provide the requested corrections and address the comments. The changes we have made in the manuscript are highlighted in red.
Point 1: The manuscript is very long and trimming details would make it easier to read and follow. A comment about the numbers in the manuscript: all numbers which should have a decimal seem to have a comma. This change needs to be made throughout.
Response 1: We trimmed the unnecessary details and replace the comma with dot.
Point 2: Introduction
- Provides a good background and appropriate statistics.
- The objectives are well mentioned.
- It might be a better idea to move the section 1.2 “sources of information” to the methods section. The section should probably end after the aims.
- The first line under “sources of information” needs a reference when the authors mention the 2021 World Bank classification.
Response 2: We made the suggested change and added the reference.
Point 3: Methods
- The section provides good definitions for the factors examined.
- How were the socio-economic factors selected? What is the rationale for selecting BMI, alcohol intake, and smoking?
- Rationale for selecting a sample size of 171 cases are given in the results section. It would be more appropriate to move it to the methods section.
Response 3: Initially, a larger number of socioeconomic factors were analysed, but after statistical evaluation, only for these three factors (BMI, alcohol intake, and smoking) we found enough data, and also proved to have a significant impact on female fertility. We have specified these aspects in the text. We explained in the Materials and Methods chapter why we analysed 171 countries.
Point 4: Results
- The authors provide a lot of detail in this section and at some places hard to follow. They should probably consider trimming this section and only include a few tables with key findings and probably include the other results in the supplementary section.
- It is difficult to follow Table 3 and not sure if it can just be mentioned in the text.
- It is not very clear why the authors have decided to show Table 4. They should consider mentioning it in the text or moving it to the supplementary section.
- Section 3.1 “Regression Models” is very detailed. Should consider trimming.
- Not sure why the authors have decided to include a correlation matrix in Table 6.
- Table 8 should be re-formatted and not sure if details about sum of squares, df, and mean square are needed.
- Table 10 might fit well in the supplementary section. The authors could probably briefly mention the findings from the multicollinearity tests.
- Not sure if Table 11 on diagnostics is needed in the main paper.
- Not clear why Table 12 is needed. The authors can mention this in the text.
- Figure 4 showing histogram and residual plots are probably not needed or can be moved to the supplementary section.
- Overall, the results section needs a lot of trimming and re-formatting. Presently, has too many details and some sections contains a lot of information. It might be better to have few tables and figures which show the key findings as it will make it easier to understand the findings.
Response 4: We removed tables 3 and 4 and included the data in the text. Tables 10, 11 and 12 were eliminated and the data were included in the text. We hope that after these changes the Results chapter will be easier to follow. We could not follow all the suggestions given, so as not to interrupt the logical flow of our analysis; we are sorry for that.
Point 5: Discussion
- This section should also be trimmed. Contains too many details.
- Should begin with a summary of findings and then move on to mentioning statistics from other countries.
- Well written section but with a little re-formatting and trimming, will become easier to follow.
Response 5: We have removed from the Discussion chapter some phrases that were not absolutely necessary, in order to be easy to follow.
Thank you again for reviewing our manuscript,
Elena Țarcă, MD, PhD
Reviewer 2 Report
In their paper, the Authors aimed to explore the linkage between fertility and income. They assembled a country-specific dataset on birth rate and socio-economic factors for 171 countries, using data integrated from publicly available data sources. By using regression models a negative factor with the greatest impact on female fertility was found considering the level of income per capita. Additionally, a negative effect of smoking, alcohol consumption and body weight on female fertility are also demonstrated, but with a lower impact compared to the average income per capita.
The study could be of interest with an old research question and an answer obtained by country-specific dataset. Limitations of the study have been listed, first of all the heterogeneous dataset from different countries.
Author Response
Response to Reviewer 2
Dear Reviewer,
Thank you very much for evaluating our manuscript. The changes we have made in the manuscript are highlighted in red.
Kind regards,
Elena Țarcă, MD, PhD
This manuscript is a resubmission of an earlier submission. The following is a list of the peer review reports and author responses from that submission.